Article 

# Transition towards plate tectonics tracked in the metamorphic signature of Neoarchean synmagmatic transpression

Ivan Zibra[1,2] ✉, L. J. Morrissey[3], M. De Paoli[1], D. Kelsey[1] & F. J. Korhonen[1]

Secular mantle cooling has progressively strengthened Earth's lithosphere, enabling a variety of tectonic styles. The emergence of transpressional orogens in the Neoarchean is interpreted to reflect this strengthening, during a transitional phase leading to plate tectonics. However, direct constraints on lithospheric strength remain limited due to the fragmentary Archean rock record and the scarcity of structurally and temporally constrained metamorphic data. Here we present metamorphic data from two major Neoarchean shear zone networks in the transpressional Yilgarn Orogen (Western Australia), showing shearing events with vertical components of displacement of ~10 km, and burial–exhumation rates comparable to those in Phanerozoic orogens. Our findings support numerical predictions of a mechanically strong Neoarchean lithosphere capable of sustaining significant orogenic thickening. This provides new constraints on the rheology of early continental lithosphere and offers insight into the geodynamic processes that preceded the full establishment of plate tectonics.

Secular mantle cooling through Earth's history has led to increased lithosphere strength, producing a variety of tectonic styles[1–4].

Although a wide variety of models exists, the prevailing view suggests that the Hadean to mid-Archean (≥3200 Ma) Earth was probably dominated by stagnant to squishy lid tectonic modes[2,5–8]. Experiments and field constraints suggest that weak lithosphere accommodated shortening through broad zones of orogen-parallel distributed flow, associated with minimal vertical displacement between adjacent blocks, and limited topographic relief[9–11]. By contrast, modern Earth is governed by plate tectonics, which includes rigid plates joined by a global network of narrow plate boundaries, along which most deformation is localized[6]. Here, stronger, stiffer lithosphere promotes strain localization along major shear zones, allowing the development of crustal thickening (or thinning, along divergent plate boundaries) and pronounced topography[12–15]. The timing of when plate tectonics was fully established and Earth's lithosphere underwent a change in tectonic style is debated[16]. However, diverse geological evidence points to the Neoarchean (2800–2500 Ma) as a critical transitional period preceding the establishment of plate tectonics. Numerical and thermal models indicate that, during this time, the lithosphere crossed a rheological threshold, facilitating strain localization, crustal thickening, and exhumation[17,18]. This transition coincided with the progressive global emergence of key tectonic features, as recorded in granite geochemistry[19], metamorphic patterns[20], and a range of geological proxies[2]. While modern-style plate tectonics likely did not emerge until the Neoproterozoic[21,22], the Neoarchean marks a pivotal stage in Earth's tectonic evolution[16].

Transpressional orogens became widespread towards the end of the Archean[23,24]. It remains contentious whether this reflects the progressive secular strengthening of continental lithosphere[18], or a peculiar tectonic style achieved during convergence of weak lithosphere[25]. Transpressional orogens may have been mainly developed along ancient lithospheric boundaries, representing a possible environment for the onset and progressive establishment of plate tectonics[26]. To advocate for the Neoarchean as a key period of transition between tectonic styles requires quantitative datasets that constrain the magnitude of burial and exhumation along shear zones.

[1]Geological Survey of Western Australia, Perth, WA, Australia. [2]School of Earth, Atmosphere and Environment, Monash University, Clayton, VIC, Australia. [3]Future Industries Institute, University of South Australia, Mawson Lakes, SA, Australia. ✉e-mail: ivan.zibra@dmpe.wa.gov.au

Such datasets are limited, and typically bear large uncertainties both in terms of timing of fabric development and overall tectonic significance[2,27,28], but need to come from structurally and time-constrained metamorphic pressure–temperature (P–T) data[29].

From a structural and metamorphic perspective, efficient strain localization in strong lithosphere allows exhumation of deep metamorphic rocks along major shear zones[12,14,30], with P–T paths characterized by trajectories indicative of fast exhumation[31,32]. In contrast, the distributed nature of strain in weak lithosphere implies that slow erosion may have been the prevailing exhumation mode for metamorphic rocks, with P–T paths approaching isobaric cooling[9,33,34]. Structural and metamorphic datasets are therefore critical tools for tracing tectonic styles through Earth's geological archive.

To address the paucity of quantitative pressure–temperature–time (P–T–t) data useful for underpinning Archean tectonic models, here we present metamorphic data from two large-scale, synmagmatic shear zone systems, with a well-constrained tectono-magmatic evolution, from the Yilgarn Orogen of Western Australia. We estimate the amount of vertical displacement recorded by these structures, shedding light on the prevailing tectonic style associated with late-Archean synmagmatic transpression.

The Superior Province (Canada) and the Yilgarn Craton (Western Australia) expose archetypal examples of late Archean, large-scale transpressional orogens, typified by crustal-scale shear zones, some of which juxtapose distinct terranes[23,35–39]. Both transpressional belts include generalized shallow-dipping seismic reflectors[24], and upper-mantle seismic reflectors that correspond to major tectonic boundaries at the surface[40,41], together with arc-like geochemical signature of volcano-sedimentary greenstone sequences[42,43], and supracrustal rocks buried to lower-crustal depths along terrane boundaries[44].

The core of the Yilgarn Craton (Fig. 1) exposes upper-crustal, Meso- to Neoarchean granite–greenstone terranes, mostly unaffected by post-Archean deformation[45,46]. In the western Youanmi Terrane (YT), supracrustal rocks of the Murchison Supergroup were produced during the 3000–2750 Ma period of prevailing lithospheric extension, punctuated by episodes of granitic diapirism[47,48]. The docking of the westernmost Narryer Terrane with the rest of the craton[40,44,46] marks the onset of the 2730–2660 Ma Yilgarn Orogeny, with regional-scale crustal thickening and granitic magmatism occurring along crustal-scale transpressional shear zones[35,49]. Rifting focused in the eastern half of the craton (in the Eastern Goldfield Superterrane, EGST) was associated with the emplacement of the 2720–2690 Ma greenstone sequence of the Kalgoorlie Group[42], with craton-wide regional shortening resuming at c. 2680 Ma, producing – or reactivating – most of the large-scale shear zones exposed today[50]. In the southwestern portion of the YT, along the Corrigin tectonic zone (CTZ, Fig. 1[51], peak granulite-facies conditions developed at 2651 ± 2 Ma[52]. The two case studies presented here (Fig. 1) are from two shear zone systems located about 250 km apart: the Cundimurra–Tuckabianna (YT) and the Ballard (EGST). Notably these two structures do not coincide with the main terrane boundaries (Fig. 1).

The structural architecture of the YT was mainly shaped via two major crustal shortening periods, during which the Yarraquin and Cundimurra plutons were emplaced along adjacent reverse shear zones, named Tuckabianna and Cundimurra shear zones (TSZ and CSZ, respectively, Figs. 1 and 2a–c[53]. The Yarraquin pluton was emplaced during the c. 2730 Ma contractional events marking the onset of the Yilgarn Orogeny[49]. The Cundimurra pluton includes two main syntectonic magmatic units, with c. 2680 Ma tonalite to monzogranite intruded by 2670 ± 6 Ma porphyritic granite[54]. A 2665 ± 9 Ma $^{40}Ar/^{39}Ar$ muscovite age from a Yarraquin-pluton-derived mylonitic gneiss, in the hangingwall of the TSZ[55], indicates that exhumation and cooling of the Yarraquin pluton to about 400 ± 50 °C occurred during shearing along the adjacent CSZ[53].

In the EGST, the regional-scale Ballard shear zone (BSZ), coeval with the CSZ, assisted the emplacement of the 2678 ± 6 Ma to 2674 ± 3 Ma synkinematic Ballard Pluton into greenstones of the Kalgoorlie Group[35]. The 2670 ± 4 Ma metamorphic age from migmatitic slivers within these sheets[35], provides the best estimation for the age of the corresponding peak assemblage in the contact aureole studied here, in the footwall of the BSZ. Shear fabrics along the BSZ developed during the regional-scale $D_1$ transpressional event, predating the emplacement of c. 2665 Ma, post-kinematic granite bodies[35].

The effects of these synmagmatic shortening events are recorded in the western portion of the craton (Fig. 1), where peak metamorphic conditions in the Yalgoo dome area (amphibolite-facies, 2685 ± 15 Ma[56] and in the Narryer Terrane (granulite-facies, 2690–2665 Ma[57]) were broadly coeval with the activity of the CSZ and the BSZ.

The two shear zone networks examined here are crustal-scale, east-dipping structures that bound north–south elongated and wedge-shaped granitic plutons (Fig. 2). Besides first-order geometry, these structures share a bulk kinematic framework represented by inclined partitioned transpression, in which synkinematic granitic sheets accommodated large amounts of strike-slip, orogen-parallel strain, while dominant flattening in stronger, amphibolite-dominated supracrustal rocks accommodated the extrusion of hot shear zone cores[35]. Although extrusion tectonics is better constrained for the 2680–2670 Ma shear zones studied here, such as the CSZ[54] and the BSZ[35], evidence of syn-emplacement contractional tectonics is widespread also for the early-orogenic Yarraquin pluton[49].

The three plutons and associated shear zones examined here also share meso- to microstructural features that are consistent with magma crystallization and cooling along active shear zones[58,59]. This includes[35,49,54] (i) concordant transition from magmatic to high-temperature solid-state fabrics, both concordant with regional-scale tectonic fabrics; (ii) widespread occurrence of melt-present shearing in both the plutons and their country rocks, and (iii) synkinematic porphyroblasts in the contact aureoles.

Notably, deformation was highly heterogeneous at the craton scale. Supracrustal country rocks were pervasively deformed only in the proximity of the synmagmatic shear zones while, away from major high-strain zones, large portions of the internal stratigraphy and most of the primary depositional/volcanic structures of the greenstone sequences are well preserved[48,60,61].

In summary, although the development of the main tectonic events in the Yilgarn Craton is well constrained from a structural, stratigraphic and geochronological viewpoint, there is a generalised lack of data about the magnitude of vertical displacement recorded by the main Yilgarn shear zones.

## Results
### Yarraquin Pluton
We studied a garnet-amphibolite (sample 199687) from a greenstone sliver exposed along the Lake shear zone (a subsidiary of the TSZ, Fig. 2a), which was active during the c. 2730 Ma syndeformational crystallization of the Yarraquin pluton[49]. The amphibolite is a migmatitic SL tectonite, with synkinematic garnet-bearing tonalitic leucosomes (Supplementary Section 1, Supplementary Fig. S1a–d) that developed during the magmatic crystallization of host tonalite[49]. The latter exhibits widespread evidence of melt-present shearing.

The peak assemblage in this sample is garnet–hornblende–plagioclase–ilmenite–quartz–melt. This assemblage occurs in a large P–T stability field between 4.5 and 10 kbar and 730–825 °C (Fig. 3a). By using modal proportion estimates from two thin sections (garnet: 6–7 mol.%; hornblende: 32–40 mol.%; plagioclase: 36–44 mol%) we refined peak P–T conditions to 5–7.5 kbar and 750–810 °C. The range in modal proportions between the two thin sections reflects the inevitable grain-scale heterogeneity of migmatitic rocks.

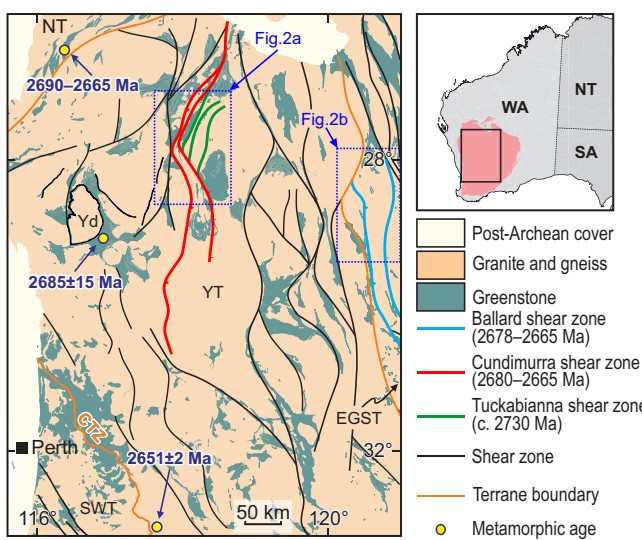

**Fig. 1 | Simplified geological map of the Yilgarn Craton.** The map shows the network of craton-scale shear zones, together with the two shear zone systems studied here, and the subdivision into main terranes (NT Narryer, YT Youanmi, SWT Southwest, EGST Eastern Goldfields Superterrane). Yd Yalgoo dome, CTZ Corrigin tectonic zone. Only the relevant metamorphic ages outside the study areas are shown here.

## Cundimurra shear zone

We selected a staurolite–andalusite schist (sample 198197) exposed along the synkinematic contact aureole of the 2680–2670 Ma Cundimurra pluton, 200 m east of the granite–greenstone boundary (Fig. 2a). Field relationships and microstructures of this sample are detailed in ref. 54, and chief microstructures are described in the Supplementary Section 1.2 (Supplementary Fig. S1e–i).

The peak assemblage is plagioclase–muscovite–andalusite–ilmenite–quartz. Prograde staurolite occurs as inclusions within synkinematic andalusite porphyroblasts (Supplementary Fig. S1g). The field that best corresponds to the peak assemblage occurs at $P \leq 3.7$ kbar and $T$ of 480–630 °C (Fig. 3b). Retrograde chloritoid and paragonite form in narrow domains along a crenulation cleavage (Supplementary Fig. S1i). This locally developed assemblage cannot be accurately modelled using the whole-rock composition, but is suggestive of an evolution dominated by cooling.

Sample 198113 is a medium-grained, protomylonitic to mylonitic biotite-bearing granite, collected in the northern portion of the Cundimurra pluton (Fig. 2a), and containing plagioclase and K-feldspar porphyroclasts (former phenocrysts of magmatic origin) that are up to 20 mm in size (Supplementary Fig. S1j). This sample is representative of the regional-scale gneissic to mylonitic fabric developed during the post-magmatic syndeformational cooling of the CSZ[54]. Microstructural data suggest that this solid-state fabric developed at ~500 °C[62].

In the studied sample, mylonitization mainly took place within the quartz- and biotite-rich matrix in between more rigid feldspar porphyroclasts. These localised domains were modelled to determine the $P$–$T$ conditions of mylonitization. The synkinematic metamorphic assemblage includes fine-grained aggregates (typically 50–250 μm in size, Supplementary Fig. S1l) of plagioclase–epidote–K-feldspar–biotite–muscovite–albite–titanite–quartz. This assemblage occurs in a relatively narrow field that extends from the edge of the diagram at 1.4 kbar and 450 °C to 5.8 kbar and 545 °C (Fig. 3c). Assuming that pressures in this sample do not exceed the peak pressures in the contact aureole, the $P$–$T$ conditions of shearing were less than 3.7 kbar and 495 ± 10 °C. These $P$–$T$ conditions of the domanial assemblage in this sample correspond to the chlorite–chloritoid–muscovite–paragonite field in sample 198197.

## Ballard shear zone

Sample 240169 is from a metasedimentary sliver interleaved with mafic–ultramafic volcanic and intrusive rocks, within the synkinematic contact aureole of the 2678–2674 Ma Ballard pluton, about 1 km west of the granite–greenstone contact (Fig. 2b). The c. 2695 Ma maximum depositional age of the sedimentary protolith[63] indicates that it represents the youngest component of the 2720–2690 Ma Kalgoorlie Group. Textbook examples of synkinematic snowball garnet and amphibole porphyroblasts developed in amphibolite, and equivalent microstructures are recorded by andalusite and staurolite in pelitic rocks, testifying to the dynamic nature of the contact metamorphism along the BSZ[35].

In sample 240169 (Supplementary Section 1.4, Supplementary Fig. S1m–t), the interpreted peak mineral assemblage quartz–plagioclase–biotite–cordierite–garnet–andalusite–staurolite–ilmenite–magnetite is stable over the $P$–$T$ range 1.7–3.5 kbar and 520–605 °C (Fig. 3d). Retrogression is limited to localized chlorite on andalusite, providing no petrological information to constrain any $P$–$T$ path.

## Discussion

Metamorphic data from the garnet amphibolite slivers within the currently exposed portion of the c. 2730 Ma Yarraquin pluton indicate that syndeformational pluton crystallization occurred at pressures of 5–7.5 kbar, corresponding to 18.5–27.8 km paleodepth (by assuming average crustal density of 2.8 g cm⁻³, Fig. 4a, which corresponds to a crustal lithostatic gradient of 3.7 km kbar⁻¹). Conversely, metamorphic data from the pelitic schist from the contact aureole of the Cundimurra pluton (in the footwall to the TSZ, Fig. 4b) show that crystallization occurred at shallower crustal levels ($P \leq 3.7$ kbar, ≤13.7 km paleodepth), during the passive exhumation of the Yarraquin pluton in the hangingwall of the CSZ. This is consistent with the 2665 ± 9 Ma ⁴⁰Ar/³⁹Ar muscovite cooling age from the Yarraquin pluton[53]. As no structures are known to have caused a differential movement between the two synmagmatic, adjacent shear zones, in the 2730–2660 Ma time span, this implies at least 4.8–14.1 km of hangingwall exhumation was achieved during the transpressional event that assisted the emplacement of the Cundimurra pluton (Fig. 4c). Since exhumation started with the c. 2680 Ma onset of syntectonic magmatism along the CSZ[54], then exhumation rates along the CSZ (over the 2680–2665 Ma time span) were 0.32 to ≥0.94 mm/yr (Fig. 4c). The metamorphic signature of the retrograde evolution of the CSZ (sample 198113, Fig. 3c) reflects syndeformational cooling of the system at a structural level comparable to that of pluton emplacement, consistent with the post-magmatic shearing evolution along the CSZ, which was dominated by strike-slip kinematics at T < 600 °C[62].

In the footwall of the BSZ, the 2695 ± 5 Ma Kalgoorlie Group metasedimentary rocks (Fig. 4d, e) record peak pressures in the range 1.7–3.5 kbar (Fig. 3d), pointing to burial to 6.7–13.0 km paleodepth (Fig. 4e, f), during the 2670 ± 4 Ma metamorphic peak that accompanied the regional-scale D₁ transpressional event[35]. Burial to peak conditions took place in 16–34 Myr, pointing to burial rates of 0.20–0.81 mm/yr. Since this metasedimentary sliver represents the youngest portion of the Kalgoorlie Group, metamorphic conditions reached by these rocks are largely due to tectonic burial, with negligible contribution from sedimentary or volcanic burial. Besides sharing similar geometry and kinematics, the two coeval shear zones studied here (CSZ and BSZ, Fig. 1) record similar amounts of vertical displacement, at comparable rates (Fig. 4c, f). Elsewhere in the craton, a minimum of 10 km of exhumation took place along the Waroonga shear zone, during the c. 2660 Ma D₂ regional transpressional event that overprinted the BSZ[64]. Here, seismic data (Fig. 2e, ref. 65) suggest that—during D₂—the northern segment of the BSZ and the rocks of the same stratigraphic level of sample 240169 experienced a second burial episode (to a current depth of ~10 km) in the footwall of the Waroonga shear zone (Fig. 2e). Amphibolite- to granulite-facies peak

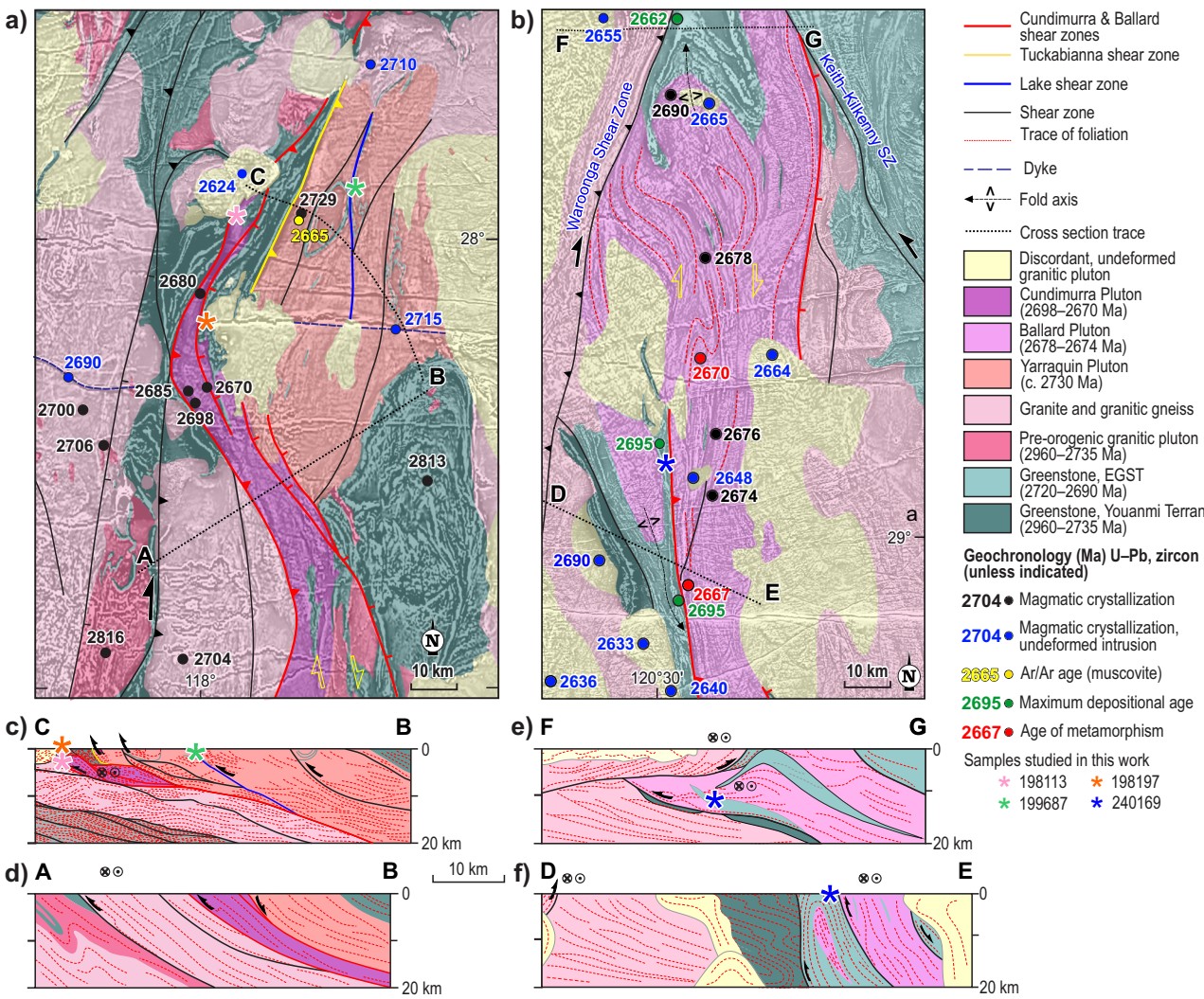

**Fig. 2 | Geological maps and cross sections of the two study areas. a** Geological map of the centra portion of the YT, in the area occupied by the Tuckabianna and Cundimurra shear zones. **b** Geological map of the western portion of the EGST, in the area exposing Ballard shear zone. **c** Cross section through the Yarraquin pluton. **d** Cross section through the Cundimurra pluton. **e** Cross section through the northern part of the Ballard shear zone. Note that rocks of the same stratigraphic level of sample 240169 experienced a second burial episode (to a current depth of -10 km) in the footwall of the Waroonga shear zone. **f** Cross section through the southern part of the Ballard shear zone. Cross sections C−B and F−G derive from the interpretation of seismic lines 10GA-YU1 and YU2, respectively [65]. Both the CSZ and the BSZ include east-dipping shear zones with opposite kinematics, resulting in the net extrusion of the shear zone core. Sample locations in cross sections C−B and F−G are projected along strike from their original location.

metamorphic conditions in the northwestern part of the craton, in the Yalgoo area and in the Narryer Terrane[56,57] were broadly coeval with the ones studied here (Fig. 1). Furthermore, along the CTZ, metasediments with 2683 ± 8 Ma depositional age reached peak granulite-facies conditions (5–7 kbar; 700–900 °C) at 2651 ± 2 Ma[52]. Although geometry, kinematic and overall tectonic evolution of the CTZ are to date poorly constrained, this structure is inferred to represent a large-scale, northeast-dipping transpressional shear zone, analogous to those studied here[51]. Similarly to the BSZ case, these P–T–t data from the CTZ indicate sediment burial to 18.5–25.9 km in a 22–42 Myr time span[52], allowing us to estimate burial rates of 0.44–1.18 mm/yr.

Tectonic overpressure and underpressure[66] may have influenced our results and paleodepth estimates. However, deviations from lithostatic pressure are difficult to quantify in the absence of independent constraints about the paleodepths reached by rock bodies, which can only be estimated in some modern orogenic settings[67].

To summarize, data from six distinct structures within the western part of the Yilgarn Craton (Fig. 1) record shearing events with

comparable amounts of vertical displacement (-10 km) and burial/exhumation rates (0.20–1.18 mm/yr), and broadly coeval metamorphic ages, suggesting craton-scale uniformity of tectonic style along major shear zones. Nevertheless, we do not exclude that other Yilgarn crustal blocks, for which P–T–t data are currently lacking, may have experienced different P–T–t trajectories, due to the intrinsically partitioned nature of the transpressional strain recorded during the Yilgarn Orogeny[35].

The rates of vertical displacement estimated here (Fig. 4c, f) are comparable to those recorded in the Superior Province[39], and along Phanerozoic melt-lubricated shear zones[68]. Together with the net amount of vertical displacement, this suggests that the Neoarchean Yilgarn lithosphere was stiff enough to allow tectonic processes at least in part analogous to those that dominate modern-style orogenic belts.

There is a perception that Archean lithosphere was inherently weak, because of the higher mantle temperatures and consequent widespread magmatism[9,69]. We suggest that—in the Yilgarn Orogen — the amount of synorogenic crustal melt that was present at any given

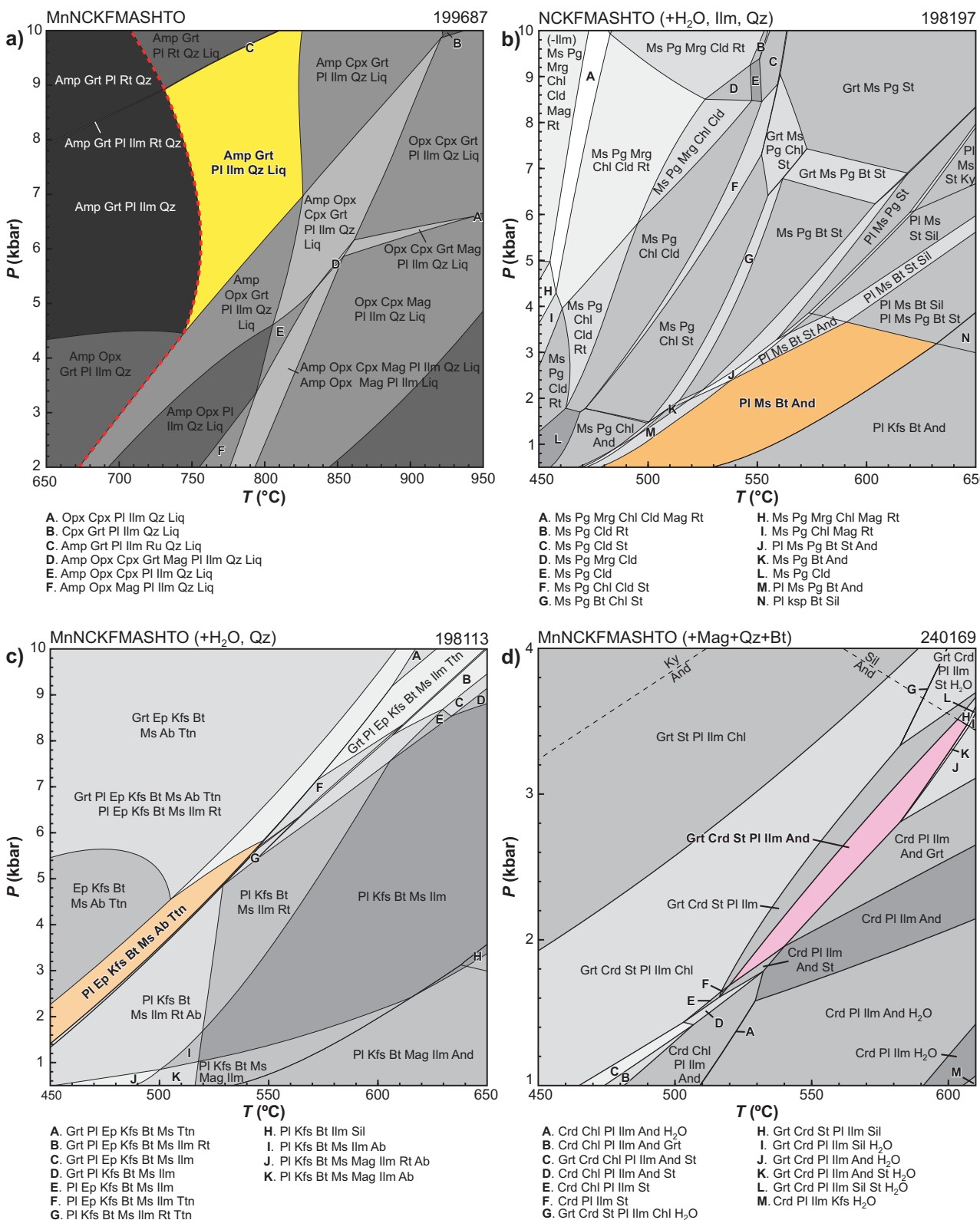

**Fig. 3 | P–T pseudosections calculated in this work. a** Sample 199687, amphibolite (YT). Darkest fields have a variance of 8. The solidus is shown as a dashed red line. **b** Sample198197, staurolite–andalusite pelitic schist. **c** Sample 198113, mylonitic granite. **d** Sample 240169, garnet–andalusite–staurolite pelitic schist. Abbreviations are after[95]. In each diagram, the peak field is highlighted in color. The modelled chemical system is provided above the diagram.

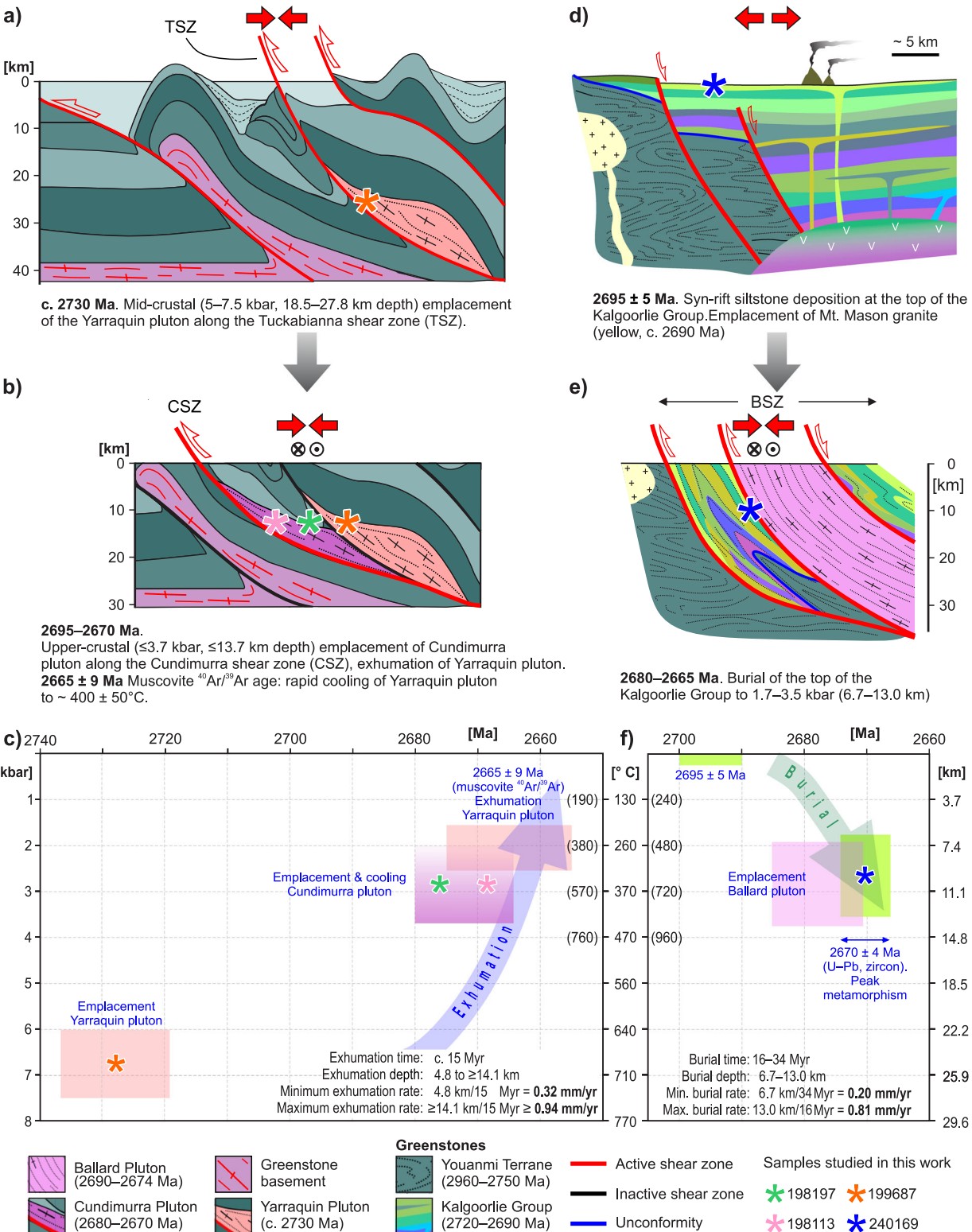

**Fig. 4 | Cartoon illustrating the tectonomagmatic and metamorphic evolution of the shear zones studied here. a** Mid-crustal emplacement of the c. 2730 Ma Yarraquin pluton during regional-scale shortening. **b** Incremental emplacement of the Cundimurra pluton causing the passive exhumation of the Yarraquin pluton. **c** Time vs. pressure diagrams for the Cundimurra and Tuckabianna shear zones area. Depth is calculated assuming average crustal density of 2.8 g/cm³. Temperature is calculated by using the average Yilgarn geothermal gradient at c. 2680 Ma[96]. Temperatures in brackets are calculated by adopting the local geothermal gradients as calculated in this work. $^{40}Ar/^{39}Ar$ muscovite age is from ref. 55. **d** Rifting event in the EGST associated with the deposition of the Kalgoorlie Group. **e** Burial of the greenstone sequence in the footwall of the BSZ, during the emplacement of the Ballard pluton. **f** Time vs. pressure diagrams for the Ballard shear zone.

time was lower than commonly perceived, and localized along high-strain channels. These channels record markedly steeper geothermal gradients than the background, average Yilgarn geotherm at c. 2680 Ma (Fig. 4c, f), because of the additional advective heat delivered along the shear zones during magmatism. These structures played therefore a key role in transferring heat upwards, extending the domain of pervasive ductile deformation to shallow crustal levels[70,71]. These channels must have experienced cycles of dramatic strain softening and high strain rates during the injection of synkinematic magma batches, due to the transient low viscosity of partially molten rocks[71]. However, our structural and metamorphic data, supported by thermal modelling results[17], indicate that the integrated strength of the Neoarchean Yilgarn lithosphere must have been relatively high, allowing efficient lateral stress propagation. In turn, this produced: (i) heterogeneous deformation at the craton scale, with pronounced strain localization along major shear zones[45]; (ii) burial/exhumation events with bulk vertical displacement in the order of 10 km or greater (Fig. 4); (iii) relatively high burial and exhumation rates (Fig. 4), which are consistent with metamorphic $P$–$T$ paths approaching isothermal decompression[44], indicative of fast exhumation; (iv) large-scale uplift and consequent erosion, with development of regional-scale unconformities[49,64].

Due to the fragmented nature of the Archean rock record, the scarcity of systematic structural and metamorphic datasets, and the presence of superimposed fabrics, direct comparison of our results (Figs. 4 and 5) with other Archean cratons remains incomplete. In the Limpopo Belt (Africa), Neoarchean tectono-metamorphic events are likely preserved, but a pervasive Proterozoic overprint obscures the geometry, kinematics and metamorphic signature of individual tectonic episodes[72]. For the North Atlantic Craton, a long-lived hot orogen, involving Meso- to Neoarchean subduction and terrane accretion has been proposed[73], though Eoarchean histories in some terranes[74] complicate the reconstruction of discrete tectonic events. Mesoarchean high-pressure, low-temperature assemblages in the Kaapvaal Craton (Africa) have been interpreted as evidence for subduction/accretion within a cold, rigid lithosphere[75], but this model is debated based on structural observations[27]. In the Pilbara Craton (Western Australia), stratigraphic, geochemical, and geochronological data suggest a Mesoarchean Wilson cycle involving rifting of a c. 3500 Ma protocraton, followed by subduction and accretion[76]. However, detailed structural studies have largely focused on the older East Pilbara Terrane, where a ~3320 Ma dome-and-keel architecture linked to granitic diapirism has been documented[77]. Both the Neoarchean Yilgarn Orogen and the Superior Province (for which more structural data are available) differ substantially, in terms of bulk geometry, kinematics and tectonic style (Fig. 5), from the style of Precambrian hot orogens, such as the Neoarchean Dharwar Craton of India, in which shortening was accommodated by broad zone of orogen-parallel distributed flow, in places associated with minimal vertical displacement between adjacent blocks, and negligible topographic relief[9,11].

Strong orogens, such as in the case of Yilgarn Craton and Superior Province, show high degrees of along-strike continuity (in map view, Fig. 1) of major structures which, in the Yilgarn case, are typically crust-penetrating and consistently east-dipping in the third dimension (Figs. 2 and 5a). Conversely, weak orogens are characterized by limited along-strike continuity of lithological boundaries and high-strain zones. In the third dimension, weak orogens show convoluted foliation patterns, which developed during prevailing vertical flow between rising granitic diapirs and sinking greenstone belts (Fig. 5b[23];). These end-member architectures are associated with distinctive structural signatures, in terms of foliation and lineation patterns, and kinematics of high-strain zones (Fig. 5c–e). Furthermore, strong transpressional orogens are associated with flattening to plane-strain steep fabrics[36–38,45], whereas

syn-convergence lateral flow in weak orogens dominantly produces flat-laying (Fig. 5d) constrictional fabrics[10,25].

Fast burial–exhumation cycles, associated with a wide range of geothermal gradients, have been both documented and modelled in Archean crustal domains characterized by granitic diapirism and greenstone sagduction[78,79]. These results highlight that neither burial/exhumation rates nor metamorphic signatures alone provide reliable proxies for ancient tectonic regimes or lithosphere rheology. Instead, it is the integration of multiple lines of evidence—including map-scale structural patterns, deformation geometries and kinematics, metamorphic assemblages, and geochemical and isotopic signatures—that enables a more robust reconstruction of lithospheric strength and tectonic environments in ancient orogenic systems. Our findings, when compared with the picture offered by weak orogens[11], support the growing consensus that a variety of tectonic styles coexisted in the Archean[80].

Convergent lines of evidence indicate that the progressive establishment of an embryonic form of plate tectonics on a planetary scale likely took place over an extended time scale, approaching and across the Archean–Proterozoic boundary[20,26]. The transition from distributed flow (in hot, weak orogens) to heterogeneous, localized high-strain zones within rigid lithospheric plates is one of the hallmarks of plate tectonics[2]. In this context, the study presented here suggests that, during the late-Archean, some relatively rigid lithospheric blocks, such as the Neoarchean Yilgarn Orogen and the Superior Province, played a significant role in focusing horizontal stresses along their boundaries, promoting lithospheric failure and some form of subduction, in the lead-up towards the establishment of modern-style plate tectonics.

## Methods
To quantify burial and exhumation along the studied synmagmatic shear zones, we constructed $P$–$T$ pseudosections (Fig. 3) for sample 199687 (Yarraquin pluton), samples 198113 and 198197 (Cundimurra pluton and its contact aureole, respectively) and sample 240169 (contact aureole of Ballard pluton).

### Whole-rock major and trace element analyses
Whole-rock major and trace elements were determined at Bureau Veritas, Perth, Western Australia. Major and minor elements (Si, Ti, Al, Cr, Fe, Mn, Mg, Ca, Sr, Ba, Na, K and P) were determined by X-ray fluorescence (XRF) spectrometry on a fused glass disk and loss on ignition (LOI) was determined by thermogravimetric analysis. The concentrations of Ag, As, Ba, Be, Bi, Cd, Ce, Co, Cr, Cs, Cu, Dy, Er, Eu, Ga, Gd, Ge, Hf, Ho, La, Lu, Nb, Nd, Ni, Pb, Pr, Rb, Sc, Sm, Sn, Sr, Ta, Tb, Th, Tl, Tm, U, V, W, Y, Yb, Zn and Zr were all determined by laser ablation inductively coupled plasma mass spectrometry (LA-ICP-MS) on a fragment of the same glass disk used earlier for XRF analysis. Data quality was monitored by blind insertion of sample duplicates, internal reference materials, and the certified reference material OREAS 24b. Bureau Veritas also included duplicate samples, certified reference materials (including OREAS 24b), and blanks. Total uncertainties for major elements are ≤1.5%, those for minor elements are <2.5% (at concentrations >0.1 wt.%) and those for most trace elements are ≤10% (Lu ±20%).

### Major element analyses and quantitative element mapping via electron probe microanalyzer (EPMA)
Chemical analyses of minerals from the YT samples were obtained using a Cameca SXFive electron microprobe at the University of Adelaide. The SXFive is equipped with five wavelength-dispersive X-ray detectors, with four utilizing large diffracting crystals. Beam conditions of 15 kV and 20 nA with a focused spot were used for all point analyses. Calibration was performed on certified synthetic and natural mineral standards from Astimex Ltd and P&H Associates. Data

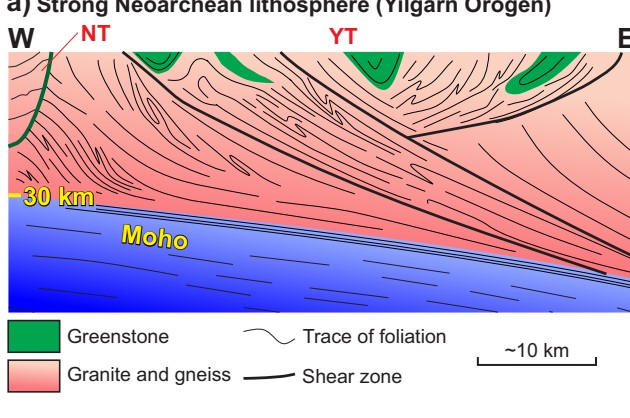

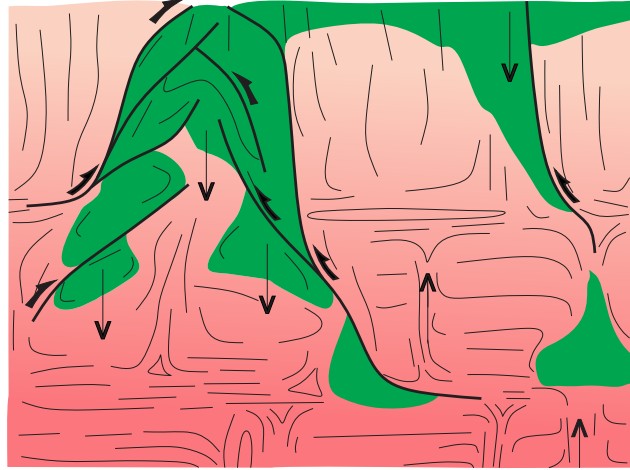

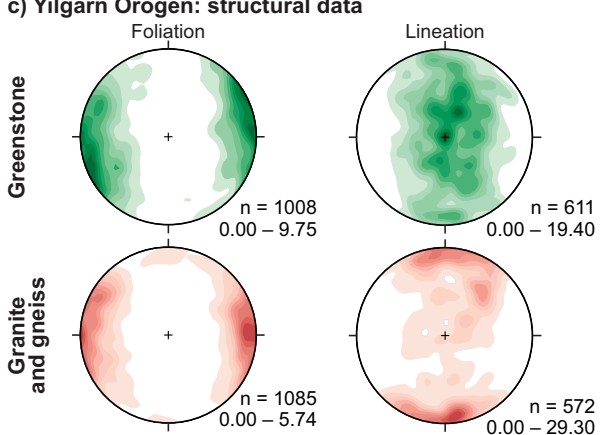

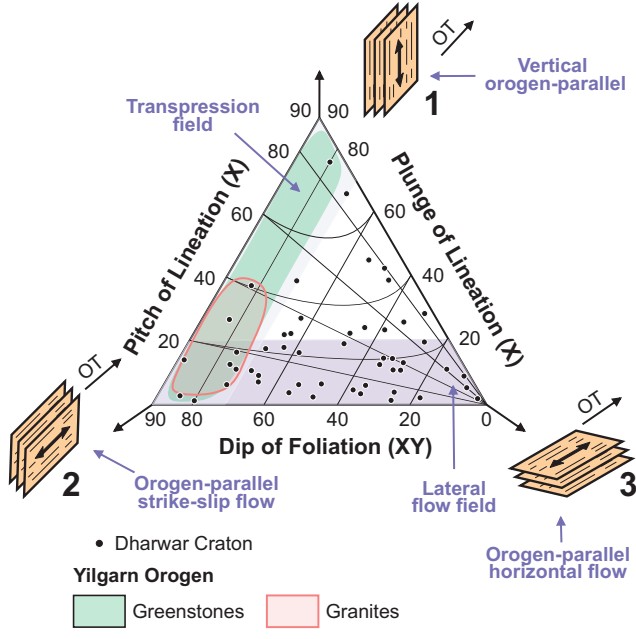

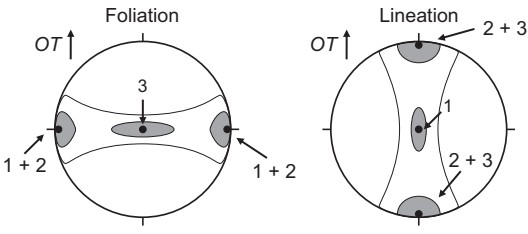

**Fig. 5 | The variety of structural patterns in end-member weak and strong Neoarchean orogens. a** Schematic cross section through the northwestern portion of the Yilgarn Craton, showing the chief crustal architecture, with east-dipping Moho and listric, crustal-scale transpressional shear zones. NT Narryer Terrane, YT Youanmi Terrane. Modified after[44]. **b** Cartoon sketching crustal geometry and kinematics in hot and weak Archean accretionary orogens, which are dominated by vertical movements (arrows), with sagduction of dense greenstones and diapiric rise of granitic plutons. Same legend as (**a**). Modified after[11]. **c** Equal-area plots showing bulk orientation of metamorphic foliation and associated stretching and mineral lineation from the main Yilgarn transpressional shear zones studied by Zibra et al.[45]. **d** Dip-pitch-plunge triangular diagram[97], here developed after[25], plotting structural data from the Dharwar (black dots) and Yilgarn (green and pink polygons) cratons, representative of the two end-members tectonic styles in Neoarchean orogens. Numbers 1–3 refer to the corresponding positions of data clusters in the plots shown in (**e**). OT: orogen trend (**e**) Stereographic projection plots of predicted distribution of foliation and lineation fabrics for the three end-member orogenic flow types shown in (**d**). Modified after[11].

calibration and reduction was carried out in Probe for EPMA, distributed by Probe Software Inc.

Mineral chemistry from sample 240169 from the EGST was acquired using a JEOL JXA-8530F Plus field emission electron microprobe (EPMA) at University of Tasmania, equipped with 5 wavelength-dispersive spectrometers. The EPMA is computer control by JEOL PC-EPMA and Probe Software Inc. "Probe For EPMA" and "Probe Image" software packages for all data acquisition and processing is used. Instrument operating conditions were 15 kV/10 nA with a 5 μm defocused beam. Matrix corrections of Armstrong-Love/Scott φ(ρz)[81] and Henke MACs were used for data reduction. Mean

Atomic Number (MAN) background correction[82,83] was used over traditional 2 point background interpolation. Well-characterized natural minerals were used as standards for microprobe analytical sessions.

## Phase equilibria modelling

Phase equilibria models for samples 198197, 199687, 240169 were based on bulk-rock composition determined by X-ray fluorescence spectroscopy, together with loss on ignition (LOI), (Supplementary Data 1). The calcium concentration of sample 240169 was corrected to account for the presence of apatite. Amphibolite sample 199687

contains alteration of plagioclase that is likely to have affected the $K_2O$ content of the sample, resulting (in the modelling) in the stabilization of biotite and K-feldspar, neither of which are present in the rock. Therefore, the $P-T$ model for sample 199687 was calculated using a $K_2O$ content derived from the proportion and electron microprobe composition of hornblende (Supplementary Data 1). Mylonitic granite sample 198113 was modelled using a domainal bulk composition calculated from maps determined by SEM-based mineral liberation analysis (i.e. TIMA) and electron microprobe compositions (Supplementary Fig. S3 and Supplementary Data 1). Bulk compositions used for modelling are provided above each model (Fig. 3).

Phase equilibria modelling was done using THERMOCALC v3.40[84]. Samples of pelite and mylonitic granite (198113, 198197 and 240169) were modelled using the internally consistent dataset ds62, of ref. 85. Sample 198197 contains negligible MnO and was modelled in the NCKFMASHTO ($Na_2O-CaO-K_2O-FeO-MgO-Al_2O_3-SiO_2-H_2O-TiO_2-Fe_2O_3$) system using the activity–composition ($a-x$) models of ref. 86, with the exception of white mica. The most up-to-date white mica model is not appropriate for sodic bulk compositions, as it predicts muscovite compositions with very high Na contents that fall within the muscovite–paragonite solvus. We therefore used a new white mica model (modified by Tim Holland, 2017, from ref. 87, with full a-x file provided as Supplementary data 2. Samples 240169 and 198113 were modelled in the MnNCKFMASHTO system using the Mn-bearing $a-x$ models of refs. 86,88. Sample 240169 used $a-x$ relationships for feldspar detailed in ref. 89.

Amphibolite sample 199678 was modelled using ds63[85] in the MnNCKFMASHTO system, using the following $a-x$ models: metabasite melt, augite and hornblende[90]; Mn-bearing garnet, orthopyroxene and biotite[86]; magnetite–spinel[91]; ilmenite–hematite[92]; plagioclase and K-feldspar[93]. The model was calculated in an MnO-bearing system as garnet and ilmenite in this sample contain appreciable MnO (3.1–5.5 wt% and 4.2–6.9 wt%, respectively). However, we acknowledge that the absence of Mn-bearing models for augite, amphibole and tonalite melt may result in a slightly increased stability field for garnet and ilmenite relative to other phases, and is a limitation of the modelling.

The oxidation state for the calculation of $P-T$ diagrams for the studied samples was based on the modal proportion and mineral chemistry of $Fe^{3+}$-bearing minerals and was refined using pseudosection exploratory calculations to ensure the models stabilized the main Fe-oxide minerals and representative peak assemblage modal proportions present within the samples (Supplementary Fig. S2b. Supplementary Data 1). Samples 198113 and 198197 were calculated with $H_2O$ in excess. The $H_2O$ content for sample 199687 was determined by combining the modal abundance of hornblende, determined from image analysis of photomicrographs, in combination with an assumed $H_2O$ content in amphibole. $H_2O$ content for sample 240169 was based on the measured amount of LOI, reduced slightly such that modal proportions of the peak assemblage were adequately represented. During preliminary calculations lower bulk $H_2O$ content increased the mode of andalusite and decreased the mode of cordierite.

Compositional and mode isopleths for all phases were calculated using the software TCInvestigator[94]. Modal proportions of each of the phases in the sample were estimated using image analysis of photomicrographs, high-resolution thin section scans or TIMA/MLA maps. Models contoured for selected isopleths are presented in Supplementary Figs. S4 and S5.

## Data availability
The data generated in this study are provided in the Supplementary Information/Source Data file, and are available in the Open Science Framework database: https://doi.org/10.17605/OSF.IO/QXJAZ.

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

## Acknowledgements

I.Z., M.D.P., D.E.K. and F.J.K. publish with the permission of the director of the Geological Survey of Western Australia. R. Quentin De Gromard is thanked for suggestions on an early version of the manuscript. Mike Prause and Deenikka Loprese are thanked for their help with preparing the figures.

## Author contributions

I.Z. designed the research project, conducted fieldwork, including mapping, sampling, microstructural and petrographic work. He wrote most of the manuscript and coordinated the contributions of the coauthors. LJM and MDP conducted the metamorphic work and contributed to the writing and the editing of the manuscript. In addition, MDP conducted part of the mapping and sampling. DEK and FJK contributed to the metamorphic work, and to the writing and editing of the manuscript.

## Competing interests

The authors declare no competing interests.
