## [Transparent Peer Review file · Nature Communications]

Transition towards plate tectonics tracked in the metamorphic signature of Neoproterozoic transpression

Corresponding Author: Dr Ivan Zibra

Version 0:

Reviewer comments:

Reviewer #1

(Remarks to the Author)

The literature suggests that the early Archean lithosphere would have been hotter, weaker, and unable to support plate tectonic style deformation. In contrast, Neoproterozoic lithosphere should have been strong enough to sustain plate tectonic style deformation. To test this, Zibra et al. investigate the Pressure-Temperature-time pathways of metamorphic samples from two major Neoproterozoic transpressional shear zones of the proto-Yilgarn craton. They found that these metamorphic samples reflect ~10 km vertical displacements for these major shear zones over ~15-20 million years, rates that are comparable with those of Phanerozoic orogens. As such, they conclude that the proto-Yilgarn craton around 2.68 Ga should have been strong enough for plate tectonics.

The manuscript is well written, and the application of TIMA/MLA, geochronology, geochemical analysis, and thermodynamic modeling approaches is classical for this kind of problem. I still have some concerns that might warrant further checks and revisions.

The authors seem to attribute all of the calculated vertical displacements to the shear zone movement. However, it is unclear to what extent the paleotopography, erosion, and sedimentary/volcanic burial could account for these values. In Fig. 4a, syn-tectonic deposition and topography are present, but these features are not shown in the rest of the figure panels. It is possible to generate km-scale basins and mountains, and they may have contributed to some kilometers of paleodepths. Given that the total displacement may be ~10km, those km-scale contributions deserve some discussion.

Similarly, emplacement of the Cundimurra Pluton below the Yarraquin Pluton and sample 199687 would have likely affected the paleodepth of this sample without taking the shear zone movement into account. This might need further investigation. Samples investigated in this work were metamorphosed and deformed in active, discrete shear zones among relatively undeformed blocks, sometimes with melt presence (e.g., migmatitic sample 199687). Under these circumstances, might tectonic overpressure or underpressure (Gerya et al., 2015; <https://doi.org/10.1111/jmg.12144>) be relevant? How might they affect pressure estimation?

Lastly, this work suggests that the early hot Precambrian lithosphere “accommodated shortening through broad zones of orogen-parallel distributed flow, associated with minimal vertical displacement between adjacent blocks, and limited topographic relief” (line 31-33). This statement is likely problematic, as significant vertical displacement and potentially topography were interpreted for early Archean terranes. For example, ref. 65 of this work shows that global >3 Ga metamorphic rocks record burial (vertical displacement) rates faster (0.32– 10 mm/yr, majorly around 1.5 mm/yr) than those interpreted from the Yilgarn samples here (0.2–0.94 mm/yr). Those rates from >3 Ga rocks were sometimes attributed to vertical tectonics (like Fig. 5e, e.g., Pilbara) and sometimes to proto-plate tectonics (e.g., Isua and Nuvvuagittuq). In Pilbara, major unconformities (e.g., between >3.41 Ga Warrawonna Group and <3.35 Ga Kelly Group) were also interpreted via diapirism (like Fig. 5e) in a supposedly hot lithosphere, indicating the presence of some topography. All of these seem hard to reconcile with the above statement made by the authors. How to resolve it?

It seems to me that the displacement rate alone may not distinguish between plate tectonics in a cold lithosphere versus (some form of) tectonics in a hot lithosphere. It is, as the authors already noted, the combination of different evidence (line 303-304) that can tell them apart. I feel the most important distinguishing features are the geometry (broad and distributed deformation zones vs. discrete zones) and kinematics (e.g., Fig. 5d), while the displacement rate may not be important. Therefore, it seems unnecessary to focus on the displacement rate (although this work did quite well in determining it). In any case, the authors should provide a better explanation of why knowing the displacement rate is important and how it can inform the evolution of the early lithosphere and tectonics.

Best regards
Jiawei Zuo

Reviewer #2

(Remarks to the Author)

Zibra et al propose a paper looking at structural and metamorphic data in the Yilgarn craton of Australia. They use the data to propose an integrated model (not so different from their previously published work), and draw comparisons with other Archean cratons.

I have no particular comments on the methodology. It is sound, and the conclusions are fully supported by the data. Others may want to propose alternative tectonic models for the Yilgarn - but the model presented here would be hard to dislodge as it is based on a wealth of field, geophysical, metamorphic and geochronological evidence. I am, in general, convinced, and I think this paper will be an important addition to the literature.

I appreciate a lot the fact that it is a paper looking at geological evidence (structural/metamorphic), evidence that is typically totally overlooked in the Archean literature focussed so much on geochemistry. It is refreshing indeed to see a description of geological features, and more to the point, it is vital and novel - not novel in terms of being new methods (for field geology is anything but), but novel in the sense that few attempt to apply such time-honoured approaches to Archean rocks. The data is good, and the interpretation is sound.

On the other hand, the evidence is a bit thin. PT paths on a few examples, while good to have, are a bit hard to extrapolate to a whole continent's story. Of course, this is what geochemists routinely do - they analyse a handful of samples and make Earth-wide inferences, so from this perspective the paper is not worse than most of the Archean literature! Still, I think the discussion section tends to get a bit lost in details and fails to extract the big picture. The most interesting portions are the ones drawing comparisons, and explaining how different cratons reacted to stresses - this is really novel and important, and is a key finding of the paper.

On balance, I think I'd support publication of this work. It is good and it is significant, and even though it is perhaps a bit thin, I think it is important that it makes its way into the literature, as it demonstrates why it is important to look at geological data. Perhaps the authors should rebalance a little bit their work, broadening the perspective and making it less local but more global. A well constrained (Yilgarn) example is key, but the real meat of the paper is the more global comparison and discussion.

JF Moyen
8/09/2025

Version 1:

Reviewer comments:

Reviewer #1

(Remarks to the Author)

The revision has addressed my concerns, and I am happy with the responses. Therefore, I recommend "accept".

This file includes a list of the comments from the two reviewers, together with our replies (the latter are in blue). At the end of this file, comments made on the word file by reviewer #2 are associated with their corresponding line (L) numbers.

REVIEWER COMMENTS

Reviewer #1 (Remarks to the Author):

The literature suggests that the early Archean lithosphere would have been hotter, weaker, and unable to support plate tectonic style deformation. In contrast, Neoproterozoic lithosphere should have been strong enough to sustain plate tectonic style deformation. To test this, Zibra et al. investigate the Pressure-Temperature-time pathways of metamorphic samples from two major Neoproterozoic transpressional shear zones of the proto-Yilgarn craton. They found that these metamorphic samples reflect ~10 km vertical displacements for these major shear zones over ~15-20 million years, rates that are comparable with those of Phanerozoic orogens. As such, they conclude that the proto-Yilgarn craton around 2.68 Ga should have been strong enough for plate tectonics.

The manuscript is well written, and the application of TIMA/MLA, geochronology, geochemical analysis, and thermodynamic modeling approaches is classical for this kind of problem. I still have some concerns that might warrant further checks and revisions.

The authors seem to attribute all of the calculated vertical displacements to the shear zone movement. However, it is unclear to what extent the paleotopography, erosion, and sedimentary/volcanic burial could account for these values. In Fig. 4a, syn-tectonic deposition and topography are present, but these features are not shown in the rest of the figure panels. It is possible to generate km-scale basins and mountains, and they may have contributed to some kilometers of paleodepths. Given that the total displacement may be ~10km, those km-scale contributions deserve some discussion.

We agree with the reviewer that some additional, clarifying words are useful when discussing this topic. At the relevant point in the discussion (near line 245) we have added the following sentence: "Since *this metasedimentary sliver represents the youngest portion of the Kalgoorlie Group, metamorphic conditions reached by these rocks are largely due to tectonic burial, with negligible contribution from sedimentary or volcanic burial.*" The effect of syn-rift paleotopography is difficult to quantify, but we infer that it would have been negligible in this case, because of the dominant effect of tectonic burial in the footwall of the shear zone.

Similarly, emplacement of the Cundimurra Pluton below the Yarraquin Pluton and sample 199687 would have likely affected the paleodepth of this sample without taking the shear zone movement into account. This might need further investigation.

We agree with the reviewer that this is – in principle - a valid observation. However, the available evidence from geological maps and cross sections (Fig. 2a and c; see also fig. 7 in Zibra et al., 2017, Tectonics, for a cross section offering greater detail) indicate

that the Cundimurra Pluton is a wedge-shaped, east-dipping, thin pluton, whose thickness abruptly decreases to zero eastward, some 10 km west from the longitude of the Lake shear zone (along which sample 199687 is located). Therefore, exhumation of this portion of the Yarraquin Pluton due to magmatic inflating during the emplacement of the Cundimurra Pluton is inferred to be negligible. This information is already given in the geological map and cross section, so we prefer not to add further clarification in the text, in order to maintain a streamlined discussion paragraph.

Samples investigated in this work were metamorphosed and deformed in active, discrete shear zones among relatively undeformed blocks, sometimes with melt presence (e.g., migmatitic sample 199687). Under these circumstances, might tectonic overpressure or underpressure (Gerya et al., 2015; <https://doi.org/10.1111/jmg.12144>) be relevant? How might they affect pressure estimation?

We agree with the reviewer that deviations from lithostatic pressure may be important, and the magnitude of which has been debated for decades in geodynamics. However, as noted by Gerya 2015, independent paleodepth estimations can only be attempted in recent orogens (e.g. European Alps), and therefore it is difficult to quantify such deviations in Archean environments. This has been briefly discussed in the discussion section (L287–290).

Lastly, this work suggests that the early hot Precambrian lithosphere “accommodated shortening through broad zones of orogen-parallel distributed flow, associated with minimal vertical displacement between adjacent blocks, and limited topographic relief” (line 31-33). This statement is likely problematic, as significant vertical displacement and potentially topography were interpreted for early Archean terranes. For example, ref. 65 of this work shows that global >3 Ga metamorphic rocks record burial (vertical displacement) rates faster (0.32– 10 mm/yr, majorly around 1.5 mm/yr) than those interpreted from the Yilgarn samples here (0.2–0.94 mm/yr). Those rates from >3 Ga rocks were sometimes attributed to vertical tectonics (like Fig. 5e, e.g., Pilbara) and sometimes to proto-plate tectonics (e.g., Isua and Nuvvuagittuq). In Pilbara, major unconformities (e.g., between >3.41 Ga Warrawonna Group and <3.35 Ga Kelly Group) were also interpreted via diapirism (like Fig. 5e) in a supposedly hot lithosphere, indicating the presence of some topography. All of these seem hard to reconcile with the above statement made by the authors. How to resolve it?

It seems to me that the displacement rate alone may not distinguish between plate tectonics in a cold lithosphere versus (some form of) tectonics in a hot lithosphere. It is, as the authors already noted, the combination of different evidence (line 303-304) that can tell them apart. I feel the most important distinguishing features are the geometry (broad and distributed deformation zones vs. discrete zones) and kinematics (e.g., Fig. 5d), while the displacement rate may not be important. Therefore, it seems unnecessary to focus on the displacement rate (although this work did quite well in determining it). In any case, the authors should provide a better explanation of why knowing the displacement rate is important and how it can inform the evolution of the early lithosphere and tectonics.

Best regards

Jiawei Zuo

We agree with the reviewer, and the relevant parts of the discussion (L368–378) have been changed to remove these contradictions. In particular, we have now put more consistently emphasis on the idea that it is the combination of several distinct datasets that allow for more solid inferences on tectonic style

Reviewer #2 (Remarks to the Author):

Zibra et al propose a paper looking at structural and metamorphic data in the Yilgarn craton of Australia. They use the data to propose an integrated model (not so different from their previously published work) and draw comparisons with other Archean cratons.

I have no particular comments on the methodology. It is sound, and the conclusions are fully supported by the data. Others may want to propose alternative tectonic models for the Yilgarn - but the model presented here would be hard to dislodge as it is based on a wealth of field, geophysics, metamorphic and geochronological evidence. I am, in general, convinced, and I think this paper will be an important addition to the literature.

I appreciate a lot the fact that it is a paper looking at geological evidence (structural/metamorphic), evidence that is typically totally overlooked in the Archean literature focussed so much on geochemistry. It is refreshing indeed to see a description of geological features, and more to the point, it is vital and novel - not novel in terms of being new methods (for field geology is anything but), but novel in the sense that few attempt to apply such time-honoured approaches to Archean rocks. The data is good, and the interpretation is sound.

On the other hand, the evidence is a bit thin. PT path on a few examples, while good to have, are a bit hard to extrapolate to a whole continent's story. Of course, this is what geochemists routinely do - they analyse a handful of samples and make Earth-wide inferences, so from this perspective the paper is not worse than most of the Archean literature! Still, I think the discussion section tends to get a bit lost in details and fails to extract the big picture. The most interesting portions are the ones drawing comparisons, and explaining how different cratons reacted to stresses - this is really novel and important, and is a key finding of the paper.

On balance, I think I'd support publication of this work. It is good and it is significant, and even though it is perhaps a bit thin, I think it is important that it makes its way into the literaure, as it demonstrates why it is important to look at geological data. Perhaps the authors should rebalance a little bit their work, broadening the perspective and making it less local but more global. A well constrained (Yilgarn) example is key, but the real meat of the paper is the more global comparison and discussion.

JF Moyen

8/09/2025

L29: I understand the constraints of a Nature (Comm) paper. However, this is really a short statement for a lot of science... secular cooling is a wide topic, so are tectonic styles, and the stiffening of the lithosphere, while likely, is another can of worms...

We agree with the reviewer that much more could be said on these topics. However, in order to keep our text streamlined, we prefer to use this introducing sentence to deliver the main message relevant here (i.e. secular lithosphere strength), and leave all the details and alternative views to the cited literature, which we have now greatly updated and expanded, with respect to the submitted version.

L30: Again, I'm prepared to believe that, but you may want to substantiate it a bit more...

Similarly to the previous point, we prefer not to expand the introduction towards aspects that are not critical to the aim of this contribution, and we prefer to offer a representative (and, again, now expanded) selection of literature focused on review papers and papers using different datasets to address the topic.

L34: Closing bracket is missing

Fixed

L36: So you do not include plate boundaries (well you do obliquely but...), and do not refer to ridges or subductions, i.e. creation/destruction of lithosphere? I'm not saying you are wrong but as you know there are other (broader/narrower) definitions of PT...

We agree with the reviewer, and we have added an essential definition of plate tectonics, which is relevant to this contribution.

L39: Again, I'm willing to accept that. However I'm pretty sure there are many models, some with different findings. So a proper review would require highlighting what is common to different models and what are the remaining areas of uncertainty....

L42: The timing in particular is debatable. Some will say this transition did not happen before the paleoproterozoic (2.1 Ga) or even the Pan-African (Stern...). Others will claim that it happened (at least regionally) as early as the middle Archean.

(reply to both L39 and L42 combined) Yes we agree with the reviewer. We have slightly modified this paragraph, with the main aim to deliver the message that, although the timing for the transition to plate tectonics is debated, there is a general consensus about the Neoproterozoic being the key transitional era during which many plate tectonics features started to appear, which is relevant for our contribution.

L56: Yes. But this is not unique to the Archean. Hot crust throughout Earth's history will do that, even in younger periods - cf. Cagnard/Gapais (e.g. Tectonics 2005, the Thompson belt), arguably the Pan-African too...

Yes we agree with the reviewer, and so we have removed the term “early-Earth”, so that the sentence now outlines the behaviour of weak lithosphere, regardless of its age.

L79: Hmm, still there are differences with other (younger) accretion-subduction complexes, lack of ophiolites etc; so you can not use this analogy without some qualification.

Yes we agree with the reviewer, and we have now removed this sentence.

L94: How do they fit in the above description, do they eg correspond to contact between «terrains», etc ?

Yes we have now specified that these two structures do not coincide with the main terrane boundaries.

L221: The first two pages of the discussion are a bit too detailed and local. I know you need to present the actual evidence, but it is a bit hard to follow the thread - what is the key point, etc.?

Yes we agree, but we think this part is critically needed, as it details which regional geological constraints we have used (besides the novel metamorphic data) to estimate vertical components of displacement and burial/exhumation rates, as summarized in figure 4.

However, to assist the reader, we have now added a wrap-up sentence at the end of this paragraph (sentence starting with: “to summarize...”), to highlight the main results, as suggested by the reviewer.

L301: Yes ! So I think this sort of more global observations and comparisons are really what is required for a Nature paper !

Yes we agree, and this portion of the Discussion section has been now expanded to offer a wider comparison with some of the best preserved Archean cratons worldwide.